# Phenotype and Treatment Options for Mesenteric Lymph Node Cavitating Syndrome in Coeliac Disease: A Case Series and Literature Review

**DOI:** 10.3390/jcm13185417

**Published:** 2024-09-12

**Authors:** Federica Marchetti, Suneil A. Raju, Nicoletta Nandi, Mohamed G. Shiha, Simon S. Cross, Benjamin Rea, Giacomo Caio, David S. Sanders

**Affiliations:** 1Department of Translational Medicine, University of Ferrara, 44121 Ferrara, Italy; caigmp@unife.it; 2Academic Unit of Gastroenterology, Sheffield Teaching Hospitals NHS Foundation Trust, Sheffield S10 2JF, UK; nicoletta.nandi@unimi.it (N.N.); mohamed.shiha1@nhs.net (M.G.S.); david.sanders1@nhs.net (D.S.S.); 3Division of Clinical Medicine, Faculty of Medicine and Population Health, The University of Sheffield Medical School, Sheffield S10 2RX, UK; simon.cross@nhs.net; 4Department of Pathophysiology and Transplantation, University of Milan, 20122 Milan, Italy; 5Department of Clinical Radiology, Sheffield Teaching Hospitals NHS Foundation Trust, Sheffield S10 2JF, UK; benjamin.rea@nhs.net

**Keywords:** coeliac disease, refractory, mesenteric lymph node cavitating syndrome, mortality

## Abstract

**Background**: There is a paucity of data on mesenteric lymph node cavitation syndrome (MLNCS), a rare condition associated with coeliac disease (CD), characterized by central necrosis within enlarged mesenteric lymph nodes. The largest case series of MLNCS was completed in 1984, (n = 6) and a poor prognosis was identified. **Methods**: A case series of all patients was conducted with MLNCS treated at the UK NHS England National Centre for Refractory Coeliac Disease between 2000 and 2023. A further literature review was conducted using PubMed and Google Scholar for patients with MLNCS and coeliac disease until 2023. **Results**: In total, there were 51 patients (6 from our case series and 45 from the literature review); 57% were female, and the mean age was 52.8 years (SD: 14.01 years). The most common presenting symptoms were weight loss (80%) and diarrhea (65%), and patients often had hyposplenism (80%). Persistent villous atrophy was present in 88% of the patients. Ten patients also had Refractory Coeliac Disease. Most of the patients (90%) were on a GFD, but the effect of this is unclear. Treatment with steroids and immunosuppressants resulted in a 40% survival rate. The overall mortality was 43%, associated with cachexia, sepsis, infectious complications, and lymphoma. **Conclusions**: MLNCS has a poor prognosis, and its diagnosis should prompt further intervention and careful follow-up. Patients commonly present with weight loss and hyposplenism should prompt further investigation. Current treatment options are inadequate and novel therapies are required.

## 1. Introduction

Mesenteric lymph node cavitation syndrome (MLNCS) is a rare condition associated with coeliac disease (CD). It was first described in 1969 by Hemet et al. as a suspected lymphoproliferative malignancy involving the mesenteric lymph nodes, presenting with severe malabsorption and evidence of severe intestinal atrophy not responding to a gluten-free diet (GFD) [1,2]. At laparoscopy, multiple jejunal cysts and hypertrophic lymph nodes were identified. In this study, a trial of steroids provided a transient clinical improvement, but the patient died within four months of presentation due to cachexia. At autopsy, neoplastic complications were excluded, and the findings confirmed the diagnosis of MLNCS with a likely long standing atrophic enteropathy. MLNCS is now characterized by central necrosis of mesenteric lymph nodes with a cystic change [3]. There are limited data on the outcomes of patients with MLNCS but a significant mortality, up to 50%, has been described [4]. Mortality is mainly related to cachexia, infectious complication, lymphoma, sepsis due to a combination of hyposplenism and malnutrition, and intestinal hemorrhages secondary to ulceration [1,5]. Historically, the diagnosis required histological confirmation of lymph nodes characterized by a central cavity with acidophilic lipid content surrounded by a rim of remnant lymph node tissue. Macroscopically, these present as hypertrophic lymph nodes with a pseudo-cystic appearance, containing a milky, creamy fluid composed of mainly fat and necrotic material [5,6,7]. More recently, MLNCS can be identified radiologically through computed tomography (CT) or magnetic resonance imaging (MRI) due to its pathognomonic appearance (Figure 1) [6].

MLNCS has only been documented in patients with coeliac disease, and in the majority of cases, these are patients with refractory coeliac disease (RCD) [3]. The significance of this rare complication in CD is unclear, but most patients have a degree of persistent duodenal villous atrophy with malabsorption-related symptoms such as diarrhea, weight loss, and abdominal pain [6]. Another frequent feature seen in patients with MLNCS is functional hyposplenism [3,5]. This can be diagnosed with a peripheral blood smear showing Howell–Jolly bodies, monocytosis, lymphocytosis, and elevated platelet counts. Confirmation can be achieved through pitted red blood cell counts or radiolabeled colloid scanning of the spleen. Furthermore, splenic atrophy can occur in patients with CD, particularly impacting the size of the marginal zone and the white pulp B-lymphocyte compartment, thought to be through an autoimmune mechanism [8]. Hyposplenism also occurs after a splenectomy and is associated with inflammatory bowel disease; therefore, it is important to consider the patient’s medical history when establishing the cause of hyposplenism [9,10]. More recently, the changes in hyposplenism have been characterized using modern imaging techniques [3,11].

All patients with coeliac disease are advised to adhere to a gluten-free diet, but the optimal treatment strategy in patients with MLNCS is unclear. Given the significant mortality in this rare condition, there is a need to identify the optimal treatment to guide management and patient consultations. The largest case series on patients with MLNCS was published in 1984 [12]. We therefore present a contemporary case series from a specialist center and literature review of patients with MLNCS to assess the effect of different treatments on the prognosis.

## 2. Materials and Methods

### 2.1. Methods for Case Series

The purpose of our study was to assess the prognosis of patients with MLNCS. All cases between 2012 and 2023 of patients treated for MLNCS were identified at the National Health Service National Centre for Refractory Coeliac Disease (Sheffield, UK). All individuals included underwent clinical assessment and had a diagnosis of CD and MLNCS confirmed radiologically (either CT or MRI) after the exclusion of other possible causes of lymphadenopathy. One patient had a lymph node biopsy confirming MLNCS. All patients diagnosed with CD had villous atrophy histologically confirmed in their duodenum, positive serological testing (IgA EmA or IgA tTG), and a supporting HLA haplotype. All adult patients with the above-described radiological appearance and a confirmed diagnosis of gluten enteropathy were identified. Response to a GFD was assessed, and other causes, both infectious and neoplastic, of lymphadenopathy were excluded, with a minimum follow-up of 1 year to assess the outcomes.

All cases were then collated together and analyzed in SPSS V27 (IBM Corp, New York, NY, USA). Statistical analysis was completed to identify differences in mortality using a logistic regression, where a *p* value < 0.05 was considered statistically significant.

### 2.2. Methods for Literature Review

We searched PubMed and Google Scholar databases, with no language restrictions up to December 2023 to identify case reports or case series of patients with MLNCS. Screening was carried out by two reviewers (FM and NN) using the titles of papers and abstracts. Any discrepancies were settled by a third reviewer (SAR). Keywords used in the searches included “Mesenteric Lymph Node Cavitation Syndrome”, “Cavitating Lymph Node”, “Cavitating Lymph Node in Refractory Celiac Disease”, and “Celiac Disease Mesenteric Lesions”. References identified in manuscripts, including review articles, were also assessed to identify relevant further cases. Once the relevant studies were identified, the full publication was reviewed. Participants in the studies were patients of any age and sex, with a previous CD diagnosis, and a MLNCS diagnosis either histologically or radiologically confirmed.

### 2.3. Data Extraction and Statistical Analysis

Data were extracted by one reviewer (FM), including manuscripts in French translated by FM. Data were extracted on age, sex, symptoms, serology, histology, imaging, adherence to GFD, and treatments for CD and MLNCS. The data were then entered into Microsoft Excel (2016).

## 3. Results

### 3.1. UK NHS England National Centre for Refractory Coeliac Disease Case Series

Table 1 shows the clinical, endoscopic, and follow-up features of patients included in our case series.

#### 3.1.1. Case Report 1

A 45- yr old woman was diagnosed with CD based on positive histology, serology (IgA tTG), and HLA DQ2 positivity. The patient reported no benefit from a GFD and a marked change in bowel habits, tending towards constipation and severe abdominal pain five years after diagnosis. A colonoscopy and capsule endoscopy (CE) were both unremarkable, but a subsequent MRI confirmed MLNCS. A CT scan from three years earlier showed the same findings. A subsequent upper gastrointestinal endoscopy showed duodenal mucosal healing on a GFD. Constipation was treated, with success, with Linaclotide and Movicol.

Two years after being diagnosed with MLNCS, a CT of the thorax, abdomen, and pelvis showed multiple stable rounded left mesenteric nodes, some with rim calcifications, and hyposplenism, all findings typical of MLNCS. A repeat gastroscopy 8 years later showed a healed duodenal mucosa. She remains under monitoring and is still alive two years after her diagnosis of MLNCS, five years after MLNCS was first present.

#### 3.1.2. Case Report 2

A 68 yr old woman presented with diarrhea and profound weight loss. That same year, she was diagnosed with CD (Marsh grade 3c at duodenal histology, positive IgA tTG, and HLA DQ2 and DQ8 heterozygosity). Since the diagnosis, she has been strictly adherent to a GFD. She also had a positive anti-smooth muscle antibody, and a liver biopsy showed evidence of inflammation with nodular regenerative hyperplasia and patchy perisinusoidal fibrosis. A chest, abdomen, and pelvis CT scan three years later demonstrated multiple cystic/necrotic central mesenteric lymph nodes, alongside splenic atrophy and hepatomegaly.

Four years later, gastroscopy was repeated, and the duodenal biopsies confirmed a diagnosis of refractory coeliac disease type 2 (RCD2), with marked villous atrophy and an increased CD3+/CD8− intraepithelial lymphocytes (IELs) ratio. A subsequent MRI confirmed MLNCS with splenic atrophy, associated with longstanding CD.

Following her diagnosis with RCD2, she started therapy with prednisolone and azathioprine, with a rapid and marked improvement in symptoms. After a couple of months, she stopped prednisolone and continued with only azathioprine. Clinically, she maintained her weight, and the diarrhea resolved.

#### 3.1.3. Case Report 3

A 49 yr old man underwent a CT scan following a right lobe pulmonary empyema that required thoracoscopy and drainage, which showed evidence of MLNCS.

During surgery, a lymph node was sampled laparoscopically, confirming MLNCS. An upper gastrointestinal endoscopy was then undertaken, confirming the diagnosis of CD, with villous atrophy, positive serology (IgA tTG, IgA, and IgG anti-gliadin), and HLA DQ2 and DQ8 heterozygosity.

One year later, the diagnosis of RCD2 was made based on the persistence of villous atrophy at histology and a monoclonal T-cell population. A CE also revealed extensive atrophy within the small bowel. He was therefore started on Budesonide (open capsule, 9 mg per day). At the subsequent endoscopy three years later, there was a persisting T-cell monoclone; however, the extent of mucosal inflammation, as evidenced by CE, had decreased from 75% to 50%. The patient died 9 years after the diagnosis from an acute exacerbation of chronic obstructive pulmonary disease (COPD).

#### 3.1.4. Case Report 4

A 52-year-old woman was diagnosed with CD, with villous atrophy and HLA DQ2 positivity. Nine years later, due to recurrent symptoms, she underwent a repeat gastroscopy that showed Marsh grade 3b, and her CT scan demonstrated radiological findings typical of MLNCS. One year later, a trial of prednisolone (30 mg) improved her symptoms; however, her liver function deteriorated, and the prednisolone was reduced to 5 mg.

Three years after the MLNCS diagnosis, duodenal biopsies revealed Marsh grade 3a and a 100% monoclone T-cell population. A CE showed villous atrophy from proximal to mid small bowel. Due to progressively worsening bilirubin, a liver biopsy was completed one year later, which suggested a T-cell lymphoma. In the following months, the patient progressively deteriorated, developing respiratory problems and orbital swelling. The latter was biopsied and showed an orbital and cerebral lymphoma and she died following disease progression.

#### 3.1.5. Case Report 5

A 78-years-old woman presented with difficulties climbing stairs, loss of coordination, left leg stiffness, polyuria, and urgency. She exhibited slurred speech with broken pursuit but no nystagmus. She also had ataxia with significant stiffness affecting the left leg and deformation of the left foot with hammer toes. She had brisk reflexes, but plantar reflexes were normal, and she had a myoclonic jerk. The overall picture was suggestive of Stiff Person Syndrome, associated with gluten ataxia.

Two years later, she was diagnosed with CD and found to have low vitamin B12 and ferritin. The diagnosis was confirmed by positive IgA tTG and a Marsh grade 3c duodenal biopsies. A CT brain scan confirmed dysfunction of the cerebellum, and she was diagnosed with celiac-related brain hyperexcitability, under the spectrum of Stiff Person Syndrome cortical myoclonus.

After starting a GFD, her antibodies improved, but never normalized, and she exhibited ongoing gastrointestinal symptoms despite being on a strict GFD. Her last biopsy, 4 years after diagnosis, showed ongoing villous atrophy (Marsh 3c) without features of RCD. Serology showed negative anti-endomisial antibody (EMA), but a positive tissue transglutaminase antibody (TTG) two times the upper limit of normal. A subsequent CT scan identified MLNCS, described as an appearance of fat/fluid levels in the mesenteric nodes and hyposplenism. Her body mass index (BMI) was 11.7 kg/m^2^. She was treated with prednisone and mycophenolate but died some weeks later due to cachexia.

#### 3.1.6. Case Report 6

A 42 yr old woman presented with unexplained weight loss, altered bowel habits with progressive diarrhea, urgency, and abdominal pain.

The colonoscopy was normal, but a CT scan was requested which suggested that there was fluid in between the bowel loops in the left upper quadrant. Five months later, a laparoscopy showed small bowel mesenteric lymphadenopathy. A murky fluid was obtained from the nodes. After a second mini laparotomy, in which a wedge of mesentery with some small bowel was excised, the pathologists excluded lymphoma. There was mesenteric fat necrosis and fibrosis in the mesentery, and a gross intraepithelial lymphoid infiltrate in the small bowel mucosa.

One year later, she was found to be HLA DQ2 homozygous, and an upper gastrointestinal endoscopy identified total villous atrophy (Marsh grade 3c) with no evidence of RCD2 at duodenal sampling, negative IgA-EMA and IgA-tTG, and a low folate. Given her presentation and endoscopic findings, CD was considered, and she started a GFD as well as folate supplementation. This resulted in an improvement in her symptoms and weight gain. Since her symptoms improved, she remains on a watch-and-wait approach.

### 3.2. Literature Review

A total of 714 unique citations were retrieved, resulting in 32 publications with 45 patients included in the literature review (Figure 2). In total, 53% were female, with a mean age at MLNCS diagnosis of 52 years (SD: 13.9 years) and a mean age at CD diagnosis of 50 years (SD: 15.8 years) (Appendix A). Serology was available for 20 patients, of which 19 were positive. The most common features in MLNCS were weight loss (82%), hyposplenism (78%), and diarrhea (67%).

The majority (89%) of patients reported following a strict GFD but symptomatic response to the GFD was variable as 30% had no improvement; 50%, partial improvement; and 20%, a total resolution of gastrointestinal symptoms.

The management of patients with MLNCS was described in 68% of cases. Where management was reported, steroids were given in 70% of cases, and 57% of these patients died, of which half were within one year of their diagnosis. A watch-and-wait approach was followed in the remaining cases (30%), with 44% of them dying (half before one year and half after one year from diagnosis). The cause of death was documented in 34% of cases and due to cachexia (33%), sepsis and infectious complications (33%), lymphoma (20%), or other causes (14%).

There was no significant difference in mortality based on gender, age at CD diagnosis or MLNCS diagnosis, serology, symptoms, duodenal biopsy Marsh grading, necrotic cancerous masses, lymphoma, hyposplenism, or GFD adherence.

## 4. Discussion

By combining our literature review and case series, we found a 43% mortality within five years in patients with MLNCS. The presence of MLNCS and RCD in those with weight loss had the highest mortality. The most common presenting symptoms were weight loss (80%) and diarrhea (65%), and patients often had hyposplenism (80%) and persistent duodenal villous atrophy (88%). Treatment with steroids and immunosuppressants resulted in a 40% survival rate. Most of the patients (90%) were on a GFD, but the effect of this on prognosis is unclear.

Overall, 49% of patients received steroids, of which 60% died. It is difficult to establish the reason why treated patients had similar or worse outcomes to those not on treatment. This may be due to a selection bias, as patients offered treatment may have been deemed more unwell by their clinicians and so treated more proactively.

Selected CT imaging (Figure 1) from our case series demonstrates the typically described imaging appearances of cavitating mesenteric lymph node syndrome with microsplenia and multiple enlarged lymph nodes that are of low attenuation, with thin enhancing and occasionally calcified rims [3,6,13]. The low attenuation is reflective of their fat or fluid content, mirroring the histological findings described earlier, with some nodes also demonstrating fat/fluid levels that are reported to be unique to Coeliac disease [14]. This myriad of imaging appearances in the context of coeliac disease parallels those reported in the literature and is diagnostic of cavitating mesenteric lymph node syndrome [3,6,13].

In our study, 18% of patients (9 out of 51) had lymphoma. There is, therefore, a significant risk of lymphoma in patients with MLNCS, as has been previously reported [1]. Therefore, it is important to assess for an underlying lymphoma, through guided biopsy or, more recently, advanced imaging techniques [15].

Given the rarity of MLNCS, it is important to have a wide differential, as similar findings can be seen in mycobacterial infections (such as tuberculosis), Whipple’s disease, lymphoma with necrosis, and necrotic metastatic malignancies in the mesenteric nodes. As part of the diagnostic assessment, these conditions should be considered [13]. 

The pathophysiology of MLNCS is unclear. One explanation is that the destruction of the intestinal mucosa observed in CD results in an increase in antigenic exposure, and thus in lymphocyte depletion within the mesenteric lymph nodes and the spleen. This causes cavitation of the lymphoid organs [1,6]. An alternative explanation is that local complement activation, due to endothelial damage induced by immune complexes resulting from repeated exposure to gluten, leads to intravascular coagulation and lymph node necrosis [1,5]. 

The treatment strategies for MLNCS are mainly related to refractory coeliac disease type 1 (RCD1) and RCD2. The use of steroids, predominantly prednisolone and budesonide, aim to induce clinical remission and mucosal recovery. A combination of steroids with immunosuppressants (azathioprine and mycophenolate) might exert better histological restoration of RCD1 and 2; however, this benefit has not been shown in MLNCS [16]. Despite symptomatic improvement in 70% of patients following a GFD, 44% still died. Therefore, clinical improvement with a GFD, whilst providing some symptomatic benefit to the patient, does not resolve their underlying pathology. There is therefore a need to establish the target of treatment. Half the patients in our case series had RCD2. Pharmacological agents, including cladribine (an adenosine nucleoside analogue) and tofacitinib (a Janus kinase inhibitor), have been trialled in patients with RCD and have demonstrated clinical and histological improvement [17,18]. Similarly, autologous haemopoietic stem cell transplantation has been used in patients who do not respond to other treatment options and shows promise [16,19]. The potential therapeutic role of these treatments in patients with MLNCS has not been explored and merits investigation.

Our study found a high mortality rate in patients with MLNCS. The causes of death were mainly related to cachexia, hemorrhagic intestinal ulceration, sepsis, and infectious complications, such as pneumopathy, infectious pericarditis, and septic shock. These patients may be at higher risk due to their functional hyposplenism [6]. Another common cause of death was lymphoma, predominantly T-cell lymphoma associated with enteropathy (EATL).

No significant mortality risk factors were identified. However, in our case series, there is a suggestion that an earlier diagnosis may have some prognostic benefits. It is difficult to establish why this may be. One possible explanation is that an earlier diagnosis allows for patients to adhere to a GFD earlier, thus reducing the degree of mucosal damage incurred.

This study had limitations. The small sample size makes it difficult to draw conclusions. However, we have reported the largest study on patients with MLNCS and included all published cases. Another limitation is that older publications do not include as much information, and therefore, some data are missing. It is difficult to mitigate this, and the results have been reported where available.

## 5. Conclusions

In summary, MLNCS is a rare complication of CD and has a poor prognosis. Patients commonly presented with weight loss, hyposplenism, persistent villous atrophy, and RCD. The identification of these patients is crucial to enable close monitoring; however, current treatment strategies are inadequate. Further research is required to identify novel therapeutic options.

## Figures and Tables

**Figure 1 jcm-13-05417-f001:**
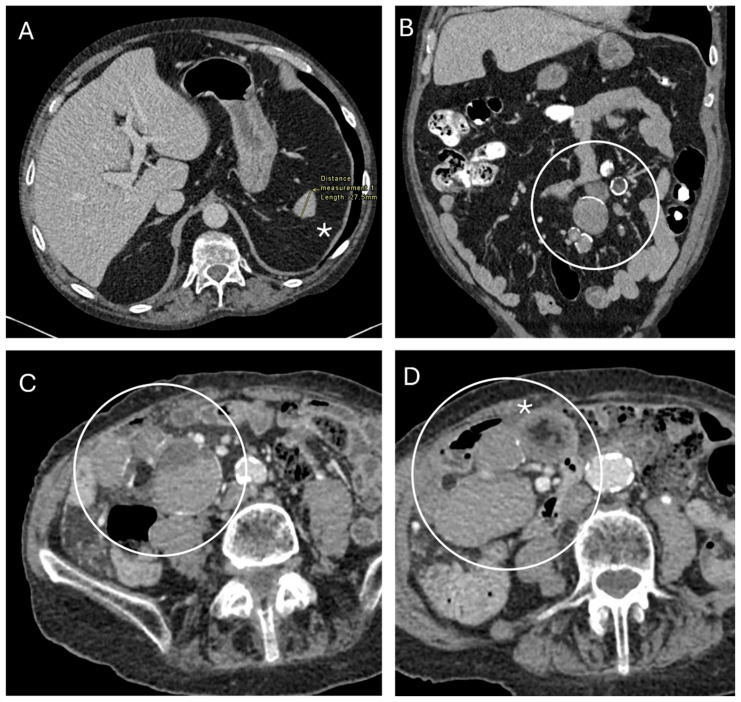
Selected images from contrast-enhanced CT scans of the abdomen and pelvis in two patients with cavitating mesenteric lymph node syndrome. (**A**) Axial CT slice demonstrates microsplenia (asterisk) with a spleen size of 27 mm. (**B**,**C**) Coronal and axial images, respectively, from the same patient demonstrate multiple mixed-density mesenteric nodules with a calcified rim (white circles). (**D**) Axial CT slice from a second patient shows multiple mesenteric nodules (white circle), one of which is mixed-density and contains fat (asterisk).

**Figure 2 jcm-13-05417-f002:**
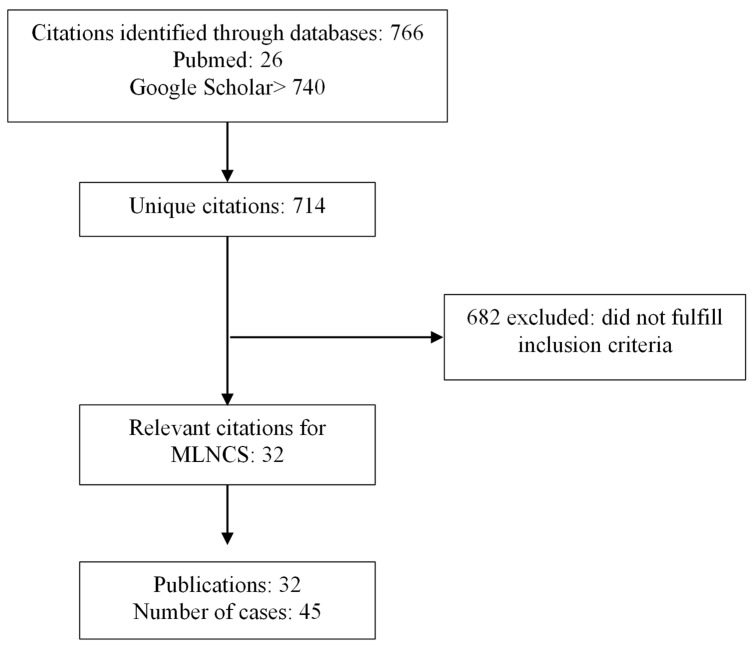
Flow diagram of the articles retrieved and inclusion progress through the literature review stages.

**Table 1 jcm-13-05417-t001:** Clinical, endoscopic, and follow-up features of patients included in our case series.

	Patient 1	Patient 2	Patient 3	Patient 4	Patient 5	Patient 6
Age of diagnosis	45 y/o	70 y/o	49 y/o	61 y/o	81 y/o	42 y/o
Sex	F	F	M	F	F	F
Coeliac disease (CD) (years since diagnosis)/Refractory Coeliac Disease (RCD) (years since diagnosis)	CD (8 years)	CD (5 years) subsequently RCD2 (1 year)	CD (9 years ago) subsequently RCD 2 (7 years)	CD (14 years) subsequently RCD 2 (12 years)	CD (6 years)	CD (4 years)
HLA genetics	DQ2+ (homozygous)	DQ2+, DQ8+	DQ2+, DQ8+	DQ2+	Not Reported (NR)	DQ2+
Presenting symptoms (for coeliac disease)	NR	Diarrhea (5 times per day), and profound weight loss	Diarrhea (8 times per day), weight loss	NR	Ataxia, cortical myoclonus	Weight loss, altered bowel habit with progressive diarrhea, abdominal pain
CD histology at diagnosis	NR	Marsh Grade 3	NR	NR	Marsh Grade 3C	Marsh grade 3c
Gluten-free diet (GFD) compliance (years)	Yes (8 years)	Yes (5 years)	Yes	Yes	No	Yes (4 yr)
Clinical response to GFD	No	Partial	Partial	NR	-	Yes
Mucosal response to GFD (years after diagnosis)	Normal mucosa (5 years)	Marsh grade 3a increased CD3+/CD8− Intraepithelial *lymphocytes (IELs)* = RCD2 (4 years)	Marsh grade 3a (3 years after diagnosis)	Marsh grade 3a (13 years after diagnosis)	Marsh grade 3c (4 years after diagnosis)	Marsh grade 3c (4 years after diagnosis)
Presentation at diagnosis for Mesenteric Lymph Node Cavitating Syndrome (MLNCS)	Abdominal pain and change in bowel habit tending towards constipation.	Rapid weight loss, bloating and distension, cholestatic liver dysfunction, persistently elevated anti-tranglutaminase antibodies (tTG) at 37.	Diarrhea, weight loss, bloating; initially presented with large empyema (drained in thoracoscopy).	Recurrent gastrointestinal (GI) symptoms.	Incidental found at computed tomography (CT) chest-abdomen performed to exclude malignancy due to worsening of respiratory function and right lower zone consolidation.	Weight loss, altered bowel habit with progressive diarrhea, urgency, and abdominal pain.
Imaging (years after diagnosis)	CT abdomen (2 years); magnetic resonance imaging (MRI) (5 years)	CT chest, abdomen and pelvis (3 years); MRI (4 years)	CT and laparoscopy (at diagnosis)	CT (9 years)	CT (4 years)	CT scan and laparotomy
CT abdomen findings	Cavitating mesenteric lymph nodes (the largest 4,3 cm), rounded left mesenteric nodes some with rim calcification.	Multiple cystic/necrotic central mesenteric lymph nodes alongside splenic atrophy and hepatomegaly.	Persisting mesenteric lymph node and hyposplenism.	Multiple cavitating nodes and microsplenia.	Appearance of fat/fluid levels in the mesenteric nodes.	Presence of fluid in between the bowel loops in the left upper outer quadrant.
MRI findings	Multiple cystic areas within the small bowel mesentery in the left upper quadrant. A few of the lesions demonstrate low signal wall on T2-weighted images. Low signal on T1.	The lymph nodal changes remain most consistent with cavitating lymph nodes syndrome. The jejunum is underdistended and not particularly well assessed.				
Laparoscopic findings			Cavitating mesenteric lymph node syndrome.			SB mesenteric lymphadenopathy, with murky fluid obtained from the nodes. Presence of fat necrosis and fibrosis, and a gross intraepithelial lymphoid infiltrate in the SB mucosa.
Serology (at MLNCS diagnosis)	Negative	Positive			Positive	Negative
Further investigation		Capsule endoscopy (CE): scalloping and mosaic pattern in the proximal small bowel (SB). Gastroscopy: partial villous atrophy in D2 and total villous atrophy in D1	Partial villous atrophy (PVA) with a clonal T-cell population at duodenal biopsies (3 years). At CE, a reduction in disease extension was observed; reduction in disease involvement from 75% to 50% following steroid treatment and GFD.	Duodenal biopsies reveal persisting monoclone of 100% in the presence of PVA. Normal colonoscopy. CE: villous atrophy from proximal to mid SB. Scalloping and atrophic mucosa proximally. Liver biopsies suggest lymphoid population with a clonal pattern which suggest T-cell lymphoma. (13 years).	Duodenal biopsy showed TVA in D1 but no features of RCD (4 years).	
Evidence of hyposplenism	Yes	Yes	Yes	Yes	Yes	Yes
Current medications for GI conditions	Linaclotide + Movicol for constipation	Prednisolone + azathioprine for RCD2.	Prednisolone, budesonide for RCD.	Budesonide, prednisolone for RCD2, Senna for constipation	Prednisolone, mycophenolate for CD and gluten-related ataxia	None
Outcome (years after MLNCS diagnosis)	Alive (6 years)	Alive (1 year)	Death (9 Years)	Death (5 years)	Death (1 year)	Alive (4 years)

## Data Availability

The raw data supporting the conclusions of this article will be made available by the authors on request.

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
