# Peer review of "Phenotype and Treatment Options for Mesenteric Lymph Node Cavitating Syndrome in Coeliac Disease: A Case Series and Literature Review"

_jcm, 2024, doi:10.3390/jcm13185417_

Round 1

Reviewer 1 Report

Comments and Suggestions for Authors

Here Dr Marchetti et al. describe 6 patients identiified and diagnosed between 2000 and 2023 as having mesenteric lymph node cavitation syndrome (MLNCS).They describe clinical features characterized by weight loss, hyposplenism and imaging (CT). Three patients died after 1, 5 and 9 years. According to the literature review, the authors found 45 patients with MLNCS, 16% of whom had refractory coeliac disease and discussed common features such as persistent villous atrophy (87%), weight loss (82%) and hyposplenism (78%) as well as the low rate of clinical response to strict gluten free diet and the poor prognosis.

There are many items that would merit important implementations to be able to argue that the syndrome found is associated with celiac disease and that the patients described had a definite diagnosis of celiac disease.
The essentials are not reported to be able to argue that these 6 patients had previously been diagnosed with celiac disease. The authors report only that the data are extracted from a database of a refractory celiac disease center. This is, however, too weak to argue that they were celiac subjects. It is known that villus atrophy is the histological hallmark of celiac disease and the refractory form, as may be hyposplenism. However, the disease is not the only condition that results in villus trophy and hyposplenism. In my opinion, the cases described here do not have sufficient evidence to support the association of this severe clinical entity with celiac disease.
The literature case series recalled by the authors (Gastroenterology 1984) describes 6 cases who, however, did not possess an HLA pattern supporting the presence of celiac disease, and other conditions should be given more consideration (doi.org/10.1111/j.1572-0241.2005.41939.x.; doi.org/10.1016/S0140-6736(10)61493-6).

Author Response

  1. There are many items that would merit important implementations to be able to argue that the syndrome found is associated with celiac disease and that the patients described had a definite diagnosis of celiac disease.
    The essentials are not reported to be able to argue that these 6 patients had previously been diagnosed with celiac disease. The authors report only that the data are extracted from a database of a refractory celiac disease center. This is, however, too weak to argue that they were celiac subjects. It is known that villus atrophy is the histological hallmark of celiac disease and the refractory form, as may be hyposplenism. However, the disease is not the only condition that results in villus trophy and hyposplenism. In my opinion, the cases described here do not have sufficient evidence to support the association of this severe clinical entity with celiac disease.

Thank you so much for this very thoughtful comment. We have clarified how the diagnosis of CD was made both in the methods and in each individual case.

*Methods:

The purpose of our study was to assess the prognosis of patients with MLNCS. All cases between 2012 and 2023 of patients treated for MLNCS were identified at the National Health Service National Centre for Refractory Coeliac Disease (Sheffield, United Kingdom). All individuals included underwent clinical assessment and had a diagnosis of CD and MLNCS confirmed radiologically (either CT or MRI), after exclusion of other possible causes of lymphadenopathy. One patient had a lymph node biopsy confirming MLNCS. All patients diagnosed with Coeliac Disease had villous atrophy at endoscopy and positive serological testing (IgA EmA or IgA tTG). HLA findings supported the diagnose of CD. All adult patients with the above-described radiological appearance and a confirmed diagnosis of gluten enteropathy were identified. Response to a GFD was assessed and other causes, both infectious and neoplastic, of lymphadenopathy were excluded with a minimum follow up of 1 year to assess the outcomes.

*Case series

Case report 1

A 45-yr-old woman was diagnosed with CD based on positive histology, serology (IgA tTG) and HLA DQ2 positive). The patient reported no benefit from a GFD and reported a marked change in her bowel habits tending towards constipation and severe abdominal pain five years after diagnosis…”.

Case report 2

“A 68-yr-old woman presented with diarrhea and profound weight loss. The same year she was diagnosed with CD (Marsh grade 3c at duodenal histology, positive IgA tTG and HLA HLA DQ2 and DQ8 heterozygosity). Since the diagnosis she has been strictly adherent to a GFD…”.

Case report 3

“A 49-yr-old man had a CT scan following a right lobe pulmonary empyema requiring thoracoscopy and drainage which showed evidence of MLNCS. During surgery, a lymph node was sampled laparoscopically confirming MLNCS. An upper gastrointestinal endoscopy was then undertaken confirming the diagnosis of CD, with villous atrophy, with villous atrophy, together with positive serology (IgA tTG, IgA and IgG anti-gliadin) and HLA DQ2 and DQ8 heterozygosity.
One year later, the diagnosis of RCD2 was made based on the persistence of villous atrophy at histology and a monoclonal T cell population…”.

Case report 4:

“A 52-year-old woman was diagnosed with CD with villous atrophy and HLA DQ2 positive. Nine years later, due to recurrent symptoms, she underwent a repeat gastroscopy that showed Marsh grade 3b, and her CT scan demonstrated radiological findings typical of MLNCS…”

Case report 5

“…Two years later, she was diagnosed with CD and found to have a low vitamin B12 and ferritin. The diagnose was confirmed by positive IgA tTG and Marsh grade 3c duodenal biopsies. A CT brain scan confirmed dysfunctioning of the cerebellum and she was diagnosed with celiac-related brain hyperexcitability, under the spectrum of Stiff Person Syndrome cortical myoclonus…”

Case report 6

“…One year later she was found to be HLA DQ2 homozygous, and an upper gastrointestinal endoscopy identified total villous atrophy (Marsh grade 3c) with no evidence of RCD2 at duodenal sampling, negative IgA-EMA and IgA-tTG, and a low folate. Given her presentation and endoscopic findings, CD was considered, and she started a GFD as well as folate supplementation…”

  1. The literature case series recalled by the authors (Gastroenterology 1984) describes 6 cases who, however, did not possess an HLA pattern supporting the presence of celiac disease, and other conditions should be given more consideration (doi.org/10.1111/j.1572-0241.2005.41939.x.; doi.org/10.1016/S0140-6736(10)61493-6).

We thank the reviewing for suggesting these excellent papers. We have added this in the introduction section as the below:

“…Another frequent feature seen in patients with MLNCS is functional hyposplenism.1,2
The spleen disorder can be suspected based on a peripheral blood smear showing Howell-Jolly bodies, monocytosis, lymphocytosis, and elevated platelet counts. Confirmation can be achieved through pitted red blood cell counts or radiolabeled colloid scanning of the spleen. Furthermore, splenic atrophy can occur in CD, particularly impacting the size of the marginal zone and the white pulp B-lymphocyte compartment, thought to be through an autoimmune mechanism.3 Hyposplenism also occurs after a splenectomy and is associated with inflammatory bowel disease and therefore it is important to consider the patient’s medical history when establishing the cause of hyposplenism4,5 The changes of hyposplenism have been characterized, more recently, using modern imaging techniques…”

Reviewer 2 Report

Comments and Suggestions for Authors

Supplementary File: The table quality and presentation must be improved. Current format is truly sub-par.

Abstract:

The abstract is written in a very disorganized manner the formatting and content need to be improved. 

Introduction:

Please mention features of hyposplenism.

Discussion:

The discussion lacks mention of radiographic features in the case series and it's consistency with published literature. Given that Radiographic modalities are important for diagnosis CMLNS, it is pertinent to include a paragraph in detail. Also discuss other pertinent diagnostic criteria.

Also expand more detail about the pathophysiology of this disease. Mention about the causes of mortality in detail.

Discuss risk of lymphoma and it's literature and pertinent incidence.

Given how rare MLNCS is, discuss the clinical importance of ruling out other differentials and mention those differentials.

Author Response

  1. Supplementary file: The table quality and presentation must be improved. Current format is truly sub-par.

We thank the reviewer for highlighting this and have significantly improved this. We think it is now much clearer to read and the format easier to follow.

  1. Abstract: The abstract is written in a very disorganized manner the formatting and content need to be improved. 

We thank the reviewer for offering the opportunity to improve the abstract. We offer you a different version which we believe is more organised.

Background –There is a paucity of data on mesenteric lymph node cavitation syndrome (MLNCS); a rare condition associated with coeliac disease (CD), characterized by central necrosis within enlarged mesenteric lymph nodes. The largest case series of MLNCS was completed in 1984 (n=6) and identified a poor prognosis.

Methods– A case series of all patients with MLNCS treated at the UK NHS England National Centre for Refractory Coeliac Disease between 2000 and 2023.

A further literature review using PubMed and Google Scholar for patients with MLNCS and coeliac disease until 2023.

Results In total there were 51 patients (6 from our case series and 45 from the literature review), 57% were female and the mean age was 52.8 years (SD:14.01 years). The most common presenting symptoms were weight loss (80%), diarrhoea (65%) and patients often had hyposplenism (80%). Persistent villous atrophy was present in 88% of the patients. Ten patients also had Refractory Coeliac Disease. Most of the patients (90%) were on a GFD, but the effect of this is unclear. Treatment with steroids and immunosuppressants resulted in a 40% survival rate. The overall mortality was 43% and associated with cachexia, sepsis and infectious complications, and lymphoma.

Conclusions – MLNCS has a poor prognosis, and its diagnosis should prompt further intervention and careful follow-up. Patients commonly present with weight loss and hyposplenism which should prompt further investigation. Current treatment options are inadequate and novel therapies are required.

  1. Introduction: Please mention features of hyposplenism.

We thank the reviewer for this important suggestion and have mentioned features as below:

“…Another frequent feature seen in patients with MLNCS is functional hyposplenism.1,2
This can be diagnosed on a peripheral blood smear showing Howell-Jolly bodies, monocytosis, lymphocytosis, and elevated platelet counts. Confirmation can be achieved through pitted red blood cell counts or radiolabeled colloid scanning of the spleen. Furthermore, splenic atrophy can occur in CD, particularly impacting the size of the marginal zone and the white pulp B-lymphocyte compartment, thought to be through an autoimmune mechanism.3 These changes have been characterized, more recently, using modern imaging techniques.1,6 …”

  1. Discussion:
    The discussion lacks mention of radiographic features in the case series and it's consistency with published literature. Given that Radiographic modalities are important for diagnosis CMLNS, it is pertinent to include a paragraph in detail. Also discuss other pertinent diagnostic criteria.

We are incredibly grateful for this comment and have added to the manuscript as below.

Selected CT imaging (Figure 1) from our case series demonstrates the typically described imaging appearances of cavitating mesenteric lymph node syndrome with microsplenia and multiple enlarged lymph nodes that are of low attenuation, with thin enhancing and occasionally calcified rims.1,7,8 The low attenuation is reflective of their fat or fluid content, mirroring the histological findings described earlier, with some nodes also demonstrating fat-fluid levels that is reported to be unique to Coeliac disease.9 This myriad of imaging appearances in the context if coeliac disease parallels that reported in the literature and is diagnostic of cavitating mesenteric lymph node syndrome.1,7,8

Also expand more detail about the pathophysiology of this disease.

We thank the reviewers for this important suggestion and have detailed the pathophysiology as below.

The pathophysiology of MLNCS is unclear. One explanation is that the destruction of the intestinal mucosa observed in CD results in an increase in antigenic exposure and thus of lymphocyte depletion within the mesenteric lymph nodes and the spleen. This causes cavitation of the lymphoid organs.7,10 An alternative explanation is that local complement activation due to endothelial damage induced by immune complexes resulting from repeated exposure to gluten, leads to intravascular coagulation and lymph node necrosis.2,10

Mention about the causes of mortality in detail.

We are grateful for this insightful suggestion and have added the below.

Our study found a high mortality in patients with MLNCS. The causes of death were mainly related to cachexia, hemorrhagic intestinal ulceration, sepsis and infectious complications, such as pneumopathy, infectious pericarditis and septic shock. These patients may be at higher risk due to their functional hyposplenism.7 Another common cause of death was lymphoma, predominantly T-cell lymphoma associated with enteropathy (EATL).

Discuss risk of lymphoma and it's literature and pertinent incidence.

We thank the reviewer for this comment and have added the below.

In our study 18% of patients (9 out of 51) had a lymphoma. There is therefore a significant risk of lymphoma in patients with MLNCS as has been previously reported.10 It is therefore important to assess for an underlying lymphoma, through guided biopsy or, more recently, advanced imaging techniques.11

Given how rare MLNCS is, discuss the clinical importance of ruling out other differentials and mention those differentials.

We are grateful for this valuable comment to strengthen the manuscript and have added the below.

Given the rarity of MLNCS, it is important to have a wide differential as similar findings can be seen in mycobacterial infections (such as Tuberculosis), Whipple’s disease, lymphoma with necrosis and necrotic metastatic malignancies in the mesenteric nodes. As part of the diagnostic assessment, these conditions should be considered. 8
